# Radiomics as a New Frontier of Imaging for Cancer Prognosis: A Narrative Review

**DOI:** 10.3390/diagnostics11101796

**Published:** 2021-09-29

**Authors:** Alfonso Reginelli, Valerio Nardone, Giuliana Giacobbe, Maria Paola Belfiore, Roberta Grassi, Ferdinando Schettino, Mariateresa Del Canto, Roberto Grassi, Salvatore Cappabianca

**Affiliations:** 1Section of Radiology and Radiotherapy, Department of Precision Medicine, University of Campania “L. Vanvitelli”, 80138 Naples, Italy; alfonso.reginelli@unicampania.it (A.R.); gi.giacobbe10@gmail.com (G.G.); mariapaolabelfiore@libero.it (M.P.B.); roberta.grassi@libero.it (R.G.); ferdinandoschettino1@libero.it (F.S.); mariateresadelcanto@libero.it (M.D.C.); roberto.grassi1@libero.it (R.G.); salvatore.cappabianca@unicampania.it (S.C.); 2Italian Society of Medical and Interventional Radiology, SIRM Foundation, 20122 Milan, Italy

**Keywords:** texture analysis, radiomics, artificial intelligence, cancer, radiogenomics, radiology

## Abstract

The evaluation of the efficacy of different therapies is of paramount importance for the patients and the clinicians in oncology, and it is usually possible by performing imaging investigations that are interpreted, taking in consideration different response evaluation criteria. In the last decade, texture analysis (TA) has been developed in order to help the radiologist to quantify and identify parameters related to tumor heterogeneity, which cannot be appreciated by the naked eye, that can be correlated with different endpoints, including cancer prognosis. The aim of this work is to analyze the impact of texture in the prediction of response and in prognosis stratification in oncology, taking into consideration different pathologies (lung cancer, breast cancer, gastric cancer, hepatic cancer, rectal cancer). Key references were derived from a PubMed query. Hand searching and clinicaltrials.gov were also used. This paper contains a narrative report and a critical discussion of radiomics approaches related to cancer prognosis in different fields of diseases.

## 1. Introduction

Surgery, radiotherapy, chemotherapy and immunotherapy represent the mainstream of anticancer therapies, that can be used either alone or in combination.

The evaluation of their efficacy is of paramount importance in oncology, and it is usually made possible by performing imaging investigations. The different imaging data sets are then evaluated by the radiologist and the response is usually classified following the response evaluation criteria [1,2].

More recently, radiologists have developed a structured report with the aim to integrate subjective considerations with quantitative and objective assessments of the extent of the lesions in order to summarize and facilitate the clinical management of cancer patients [3,4,5].

In the last decade, a new technique, called texture analysis (TA), has been developed in order to help the radiologist to quantify and identify parameters related to tumor heterogeneity which cannot be appreciated by the naked eye [6,7,8]. This analysis can use different mathematical models that are able to extract quantitative parameters from regions of interest (ROIs) of selected volumes, that are called texture features [9,10,11,12]. TA workflow consists of different processes, such as the acquisition of the imaging, delineation of the ROIs, extraction of features and statistical correlation with different endpoints, and its use is being investigated in several fields [13,14,15,16,17,18,19,20].

More recently, a different approach of dynamic TA has been developed, that analyses the variations in TA features in subsequent imaging evaluations. This approach is usually called delta texture analysis or delta radiomics (D-TA) [21,22,23,24].

With this method, it is possible to investigate the role of TA variations after therapy (usually chemotherapy or radiotherapy) or shortly after the beginning of therapy.

Both TA and D-TA features can provide imaging biomarkers that can be used to discriminate the prognosis (prognostic parameters) and/or to predict the response to specific therapies (predictive parameters).

Herein, we will discuss the impact of TA in the prediction of response and in the prognosis stratification of different anticancer strategies, focusing on breast cancer, lung cancer, gastric cancer, liver cancer and rectal cancer, as these diseases were better characterized in TA literature, and we will provide a summary of other cancer TA. Following a literature search, we will provide a narrative overview of these topics.

## 2. Texture Analysis and Prognosis—Focus on Lung Cancer

Lung cancer represents one of the most common malignancy and the leading cause of cancer death [25]. Surgery or stereotactic radiotherapy (for unfit patients) are recommended for patients at the early stage of disease, whereas a combination of different strategies, including surgery, radiotherapy, chemotherapy and immunotherapy is used in locally advanced or metastatic diseases [26,27].

Medical imaging is of paramount importance in all the phases of the clinical management of lung cancer patients, either in diagnosis, during the different therapies in order to evaluate their efficacy and in the follow up [28,29,30,31,32,33,34].

TA is usually performed on computed tomography (CT) and positron emission tomography (PET), as these techniques have been widely used for lung cancer patients [35,36,37,38,39,40,41], whereas US and MRI are not currently used in the clinical management of lung cancer patients.

CT provides anatomical characteristics of the lung cancer lesions and the surrounding organs, whereas PET can provide molecular and metabolic information on the same structures.

We will focus on TA endpoints that include tumor response assessments and prognosis prediction.

Tumor response assessment in lung cancer is usually based on RECIST criteria [1] and it is divided into four classes: complete response, partial response, stable disease and progression of disease. Cook et al. showed that different parameters in TA analysis, based on PET, such as contrast, coarseness and busyness showed good statistical correlation with RECIST evaluation and outperformed SUV based parameters after chemoradiotherapy [42]. Dong et al., conversely, showed that early changes in texture features showed a higher specificity and sensitivity to classical parameters, such as coefficient of variation and MTV [43]. CT-based TA has been applied in the prediction of pathological response by Coroller et al. [44], who analyzed a cohort of 127 locally advanced lung cancer patients undergoing neoadjuvant chemoradiation before surgery. The authors found that tumors not responsive to chemoradiation had a rounder shape and a heterogeneous texture. In another investigation, the same authors observed that [45] TA, calculated on lymphonodes, had higher correlation with residual disease than TA obtained from the primary tumor.

For prognosis prediction, both overall survival and progression-free survival analyses are included.

Different PET-based TA parameters have been associated with overall survival, such as higher contrast [46], higher tumor heterogeneity [47] and higher tumor dissimilarity [48].

As above, D-TA features have been tested in this setting with promising results [49].

Similarly, radiomics features calculated on CT, such as shape, intensity and texture features, can improve the prognosis stratification when combined with conventional clinical features, such as age, sex, performance status [50,51].

Paul et al. investigated dynamic parameters such as reduction in mean Hounsfield Unit (HU) obtained through the course of radiotherapy (D-TA). They were significantly related to overall survival [52].

Other authors have investigated the prediction either locoregional or distant recurrence after treatment.

Pyka et al. investigated the correlation of TA features in lung cancer patients treated with radiotherapy and found that entropy and correlation outperformed SUV metrics in the prediction of recurrences [53].

Mattonen et al., contrarily, compared the radiologist and the TA features in the correct prediction of recurrence after stereotactic radiotherapy, and found that TA features had a lower classification error rate (24% versus 35%) [54].

The same authors applied the TA analysis to the penumbra region—that is, the region that extended outward 10 mm from the tumor surface. The authors discovered a significant correlation with recurrence prediction and highlight the analysis of the peritumoral region which, in the future, will be correlated with the immunotherapy response [55].

Krarup et al., on the other hand, tried to validate promising TA features calculated on PET in patients undergoing chemoradiotherapy for locally advanced lung cancer and found that the pre-selected TA features were not significantly correlated with PFS [56]. This negative study is noteworthy as it highlights the importance of the variations in technical parameters and the concerns for stability and reliability of TA features for their use in the clinical setting.

Finally, it is of great interest to report two meta-analyses recently published in the field of prognosis and lung cancer. The analysis by van Laar et al. [57] concluded that the only factors affecting survival in stage III lung cancer are tumor-size and nodal-size related factors. Kothari et al. [58], again, analyzed the prognostic value of the radiomics model in lung cancer and found significant heterogeneity among the studies (I^2^ = 70.3%).

Specifically, Van Laar [57] selected 11,996 results (including 519 duplicates) for his meta-analysis and after cross-reference searching, included a total of 65 publications, with 26 individual CT-related prognostic factors for OS of patients with stage III NSCLC described. The results show that tumor diameter is prognostic for patients with stage IIIB NSCLC. In contrast, with regard to tumor volume, the included data proved too heterogeneous to draw definitive conclusions about tumor volume as a prognostic factor for this subset of patients.

In their meta-analysis, conversely, Kothari et al. [58] selected 2747 articles identifying 55 data sets and 6223 patients that were then included. Significant heterogeneity in the methodology used for feature selection and model development was demonstrated in this study. Twenty-six datasets measured the performance of radiomics-based models in predicting OS using a C index that ranged from 0.34 to 0.86. Seven data sets used the AUC, which ranged from 0.69 to 0.96. Twelve data sets with a Tripod analysis type of 2a or higher reported both a C-index and a 95% CI (or standard deviation or standard error) and were included in the meta-analysis. The random-effects estimate was 0.57 (95% CI 0.53 to 0.62). There is significant heterogeneity (I^2^ = 70.3%).

In conclusion, TA in lung cancer has, to date, demonstrated modest prognostic capabilities. Future research should aim at optimizing and standardizing TA features, work on feature selection and model development, in order to improve this approach. Currently, several prospective observational studies are accruing lung cancer patients with different endpoints, including the development of radiomics models based on CT images to diagnose malignant nodules early. These models are able to discriminate the different types of lung cancer, correlate imaging to genetic and biomolecular characterization and stratify the prognosis of lung cancer patients (see Table 1).

## 3. Texture Analysis and Prognosis—Focus on Breast Cancer

Breast cancer represents the most common malignancy in women [25], and early detection with mammography screening has been shown to have a great impact on survival [59,60]. While ultrasonography (US) is widely used for screening purposes, MRI in recent years is increasingly being used for high-risk women, as well as for staging, assessing curative effect and monitoring recurrence [61,62,63].

Breast cancer prognosis relies on immunohistochemical biomarkers, including estrogen receptor (ER), progesterone receptor (PR), human epidermal growth factor receptor 2 (HER2) and Ki-67 as substitutive molecular subtype [64,65].

In breast cancer research, the use of radiomics combined with multiple imaging modalities, clinical information and machine learning methods are under investigation, not only to detect malignant lesions and discriminating tumor grade, but also for identifying prognostic factors; for instance, the response to neoadjuvant chemotherapy (NAC) as well as the risk of tumor recurrence [66], similar to other settings [67,68,69,70,71].

Following the aim of the review, we will focus on papers dealing with TA correlation with prognosis, the assessment of responses to therapies and the prediction of recurrences, using the imaging modalities mainly used in breast cancer clinical management (US and MRI).

TA is able to offer large potential data to define the biological features of tumors for precision medicine [72]. Radiogenomics represents a specific evolution of radiomics that uses imaging capabilities to non-invasively identify or predict tumor-specific genomic alterations [73].

The biopsy of the suspected breast lesion is today the gold standard for the characterization of breast cancer; however, it evaluates only the sampled section of a heterogeneous tumor [74] and is currently not repeated for recurrent tumors. Radiogenomics can potentially evaluate the entire tumor load with the possibility of providing a non-invasive diagnosis and to closely monitor the characteristics of the lesions [75,76,77,78,79]. The combination of quantitative radiomics features with histological, clinical, and genomic data may represent a valid possibility for clinicians to develop patient-centered treatments [72,80,81,82].

Holli et al. [83], as well as Waugh et al. [84], found a correlation with entropy calculated on MRI with the differentiation between lobular and ductal carcinoma.

Other authors have investigated the potential to predict the molecular subtypes of breast cancer [85,86,87] or with tumor grade [88,89].

Braman et al. developed an MRI-based model that was able to identify different subtypes of HER2+ breast cancer patients [90]. This model could also predict response to neoadjuvant HER2 targeted therapy. It is important to note that molecular characteristics of breast cancer can be modified under the pressure of several factors such as systemic therapies, radiotherapy and so on [91,92]. Despite that, the patients usually do not repeat biopsies in the clinical management of recurrent disease, so that TA analysis, based on non-invasive imaging, such as MRI, CT can be useful to measure these molecular characteristics and to tailor the therapeutic strategies.

Other investigators have analyzed the risk of positive sentinel or axillary lymph nodes with different approaches and accuracies [93,94,95].

More recently, Chai et al. [96] used 3T DCE-MRI to extract TA features that showed an accuracy of 0.86 and an AUC of 0.91 for the prediction of axillary nodes. Liu et al. [97], again, performed a prospective study investigating the use of different models to predict the same endpoint and found that the model combining clinical information and TA parameters showed the best AUC (0.763).

Zheng et al. also [98] used deep learning radiomics with conventional US and shear wave elastography to predict the presence of axillary metastasis (AUC 0.902) and to discriminate between low and heavy burden of axillary disease (AUC 0.905). According to these studies, in the near future TA could support clinical decision-making, avoiding invasive procedures to the axilla.

Another field of investigation is the assessment of the response to neoadjuvant chemotherapy, which represents the most employed pre-operative strategy for breast cancer patients, with about half of the patients achieving a complete pathological response (pCR).

In this setting, several authors have correlated TA features calculated on MRI to predict pCR [99]. Braman et al. [100] used DCE-MRI imaging extracted from both tumoral and peritumoral regions, obtaining an AUC of 78% for the training dataset and 74% for the validation dataset. Drukker et al. [101], obtained similar prediction results of pCR (82%) and lymphonodal status (72%), with the same imaging technique (DCE-MRI).

Parikh et al. [102], contrarily, compared TA calculated on T2-weighted and contrast-enhanced T1, weighted to assess pCR, and found that T2-w showed higher sensitivity (87.5% vs. 50%). Kim et al. [103] also used both DCE-MRI and DCE-Ultrasonography and found that both methods showed good sensitivity for pCR (respectively, 81% versus 71%).

Finally, TA has been tested in the prediction of recurrences with several approaches, mostly with MRI imaging. Huang et al. [95] have used both MRI and PET/CT TA and found an AUC of 75% and 68%, respectively, for disease-free survival (DFS) at 1 and 2 years. Park et al. [104] also found that the inclusion of TA features on MRI improved the DFS estimation on Cox analysis.

Li et al., contrarily, found that TA features calculated on DCE-MRI [105] were significantly associated with several clinical, histopathologic and genomic data and were able to discriminate between patients with good and poor prognosis, with an AUC of 88%.

In conclusion radiomics in breast cancer has mainly focused on early identification of prognostic factors such as response to neo-adjuvant chemoradiation therapy (100–105) and monitoring of recurrence [61,62,63].

Specifically, one study [100] investigated a novel combined intratumoral and peritumoral radiomic approach for pCR prediction; it combined textural features extracted from a tumor and its immediate environment using routine DCE-MRI. Indeed, peritumoral radiomics have been shown to contribute to the successful prediction of pCR from pretreatment imaging. Furthermore, it has been found that the radiomic features most predictive of response appear to vary as a function of tumor molecular subtype.

In another study [90], a combination of peritumoral and intratumoral features appears to identify intrinsic molecular subtypes of HER2+ breast cancers from imaging, offering insights into the immune response in the peritumoral environment and suggesting the potential benefit for treatment guidance.

Correlations between radiomics and histological type [83,84,85,86,87], as well as tumor grade [88,89], have also been described. In particular the two most discriminating texture parameters for differentiating luminal A and luminal B subtypes proved to be sum entropy and sum variance (*p* = 0.003) with AUCs of 0.828 for sum entropy (*p* = 0.004), 0.833 for sum variance (*p* = 0.003), and 0.878 for the model combining sum entropy and sum variance texture features (*p* = 0.001). The sum entropy and sum variance showed a positive correlation with a higher Ki-67 index [83]. 

Finally, in another study [89], a radiomic model was developed to predict the Ki-67 proliferation index in patients with invasive ductal breast cancer by preoperative magnetic resonance imaging (MRI).

In this study, quantitative imaging features (n = 1029) were extracted from ADC maps and 11 features were selected to build the model that showed areas under ROC values (AUC) of 0.75 ± 0.08, accuracy of 0.71 in the training set and 0.72, 0.70 in the test set.

In conclusion, in breast cancer, despite the fact that no meta-analyses investigating the overall impact of radiomics have been published, the same pitfalls of this method still exist, with a limited reliability of texture parameters and mainly mono-institutional studies published. At the same time, two observational prospective studies are under development, respectively, to validate metastatic risk based on radiomic features following primary therapy and to assess the performance of CT-based radiomics in evaluating the response and predicting the pCR of metastatic lymph nodes after neoadjuvant therapy (see Table 1).

## 4. Texture Analysis and Prognosis—Focus on Gastric Cancer

Despite the fact that stomach cancer is one of the “big killers” with about 1,000,000 new cases per year, and the third place for mortality among cancer patients [25], there are not many studies of radiomics approaches in this field.

The main topics of radiomics research in this disease deal with differential diagnosis, prognosis assessment and response evaluation. Following the aim of the review, we will focus on prognosis assessment and response evaluation.

Several studies have been performed on the staging of gastric cancer, in order to evaluate gastric wall infiltration in the distinction between T2 and T3/4 lesions 4 or serosal infiltration for the discrimination between T3 and T4a lesions [106].

Lymph node involvement is one of the parameters with the greatest impact on clinical decisions and patient survival. Therefore, several retrospective studies aim at the creation and validation of models or algorithms that are able to predict the degree of lymph node involvement in the preoperative phase [107,108,109,110,111,112,113,114,115].

Jiang et al., on 1689 patients, developed a “radiomics signature” that significantly correlates with lymph node metastatic involvement [114]; other researchers have used machine learning [107] or deep learning [109] algorithms, retrospectively analyzing large cohorts of patients for the detection of lymph node metastases.

Many studies have focused on the role that radiomics may have in guiding the therapeutic choice in patients with stomach cancer [116] and therefore in risk stratification and response evaluation before and after medical or surgical treatment. [117,118,119,120,121,122].

Jing-Wen et al. used a deep learning software for semi-automated segmentation and developed a CT-radiomic approach to predict response to chemotherapy in patients with advanced adenocarcinoma [123].

Jiang et al. conducted a study on 1591 patients by developing radiomics signatures that can predict survival and response to chemotherapy [124].

Li-Whuchao et al. studied the value of TA features in predicting survival after radical surgery [125].

Another group developed a computational approach by integrating large-scale imaging factors, especially radiomic features at contrast-enhanced computed tomography, to predict AHS (adverse histopathological status) and clinical outcomes of patients with GC [126].

Zhang L. et al. [127] developed a radiomic model that is able to distinguish between advanced and non-advanced GIST by retrospectively analyzing 366 patients with suspected GIST.

Zhou et al., instead, studied PET/CT features contributing to prognosis prediction in primary gastric diffuse large B-cell lymphoma (PG-DLBCL) patients [128].

Other investigators have used a D-TA approach to predict the response to neoadjuvant chemotherapy in resectable locally advanced gastric cancer and found that the TA parameter GLCM-contrast was able to predict complete pathologic response with an AUC of 0.763 [129]. Finally, innovative approaches have used radiomics for a non-invasive assessment of the immune microenvironment, correlating the TA features with the Treg cell infiltration or the HER2 expression [130,131].

In conclusion, the development of radiomic models have been shown to have a good predictive performance of response to neoadjuvant chemotherapy.

In particular in one study [118], the rad_score (in the validation cohort) demonstrated a good predicting performance in treatment response to the neoadjuvant chemotherapy (AUC [95% CI] = 0.82 [0.67, 0.98]), which was better than the clinical score (based on pre-operative clinical variables)

In another study [117] otherwise a radiomics-based model that incorporated radiomic signature, serum CA72-4, and CT reported lymph node status showed good calibration and discrimination in the training cohort [AUC, 0.92; 95% confidence interval (CI), [0.89–0.95] and the validation cohort (AUC 0.86; 95% CI, 0.81–0.91).

Another study [119], instead, developed a radiomic signature using Support Vector Machine (SVM)-based methods that was shown to be an independent predictor of DFS and response to therapy.

Finally, several studies have focused on the possibility of predicting LNM in gastric cancer. The analyzed nomogram composed of radiomic scores showed, in all studies, excellent discrimination in the training and test cohorts with AUC ranging from 0.824 to 0.886 and 0.764 to 0.8456, respectively [107,108,109,110,111,112,113].

Furthermore, these models when supplemented with clinicopathological information improved their predictive ability.

In conclusion, in gastric cancer, no meta-analyses investigating the overall impact of radiomics have been published and the same pitfalls of this method still exist, as reported above. Prospective observational studies are yet to be designed; therefore, in the future, several efforts must be made for the clinical management of this disease.

## 5. Texture Analysis and Prognosis—Focus on Liver Cancer

Hepatic lesions are extremely frequent in oncology. Liver is one of the main sites of metastasis [25] and hepatocellular carcinoma (HCC) is the most common primary tumor, representing the second leading cause of death in oncological patients and the first in patients affected by cirrhosis [25,132]. In recent years, many studies have been looking for possible applications of radiomics in the study of hepatic lesions [133,134,135,136]. Nowadays, many applications have been consolidated: from early diagnosis to post-treatment evaluation and prognosis predictions [137]. Following the aim of the review, we will focus on response evaluation and prognosis correlation in both primary and metastatic hepatic cancers.

In the setting of primary hepatic cancers, several studies have been published in the field of radiomics. In one work, a radiomic model, based on CT, was used to predict the risk of recurrence in patients with early stage HC [138]. In a further study, a combined radiomic model based on MRI was designed to predict the 5-year survival of patients with HCC. The study showed that AFP, ferritin, and CEA in preoperative, macrovascular invasion, tumor size, sex, and some radiomic features, such as correlation, inverse difference moment, cluster prominence, uniformity, and GLCM energy were independently associated with the postoperative OS of patients with primary liver cancer. Therefore, these characteristics can be considered as potential imaging biomarkers for the postoperative OS of primary liver cancer [139].

Another paper investigated and integrated radiomic features with preoperative AFP and AST values in order to stratify HCC patients into risk groups prior to surgery and thus guide treatment decisions [140].

Most works investigating the potential of radiomics as a prognostic factor analyze features extracted exclusively from the intratumoural portion. On the other hand, Zhang et al. tested a radiomic model to assess the survival of patients with hepatocarcinoma after surgery. In this case, three different ROIs were placed: in the lesion, in the penumbra zone (defined as the area of liver parenchyma surrounding the lesion to a thickness of 1 cm) and in the context of healthy liver parenchyma. Changes in radiomic features from the liver parenchyma have been shown to be predictive of patient survival and may provide prognostic information regarding recurrence and metastatic potential; radiomic features involving several regions, therefore, have greater prognostic power than a single lesion assessment [141]. Song et al. also performed a retrospective study, in which radiomic features, based on contrast-enhanced MRI extracted from both the intra- and peri-tumor area, were analyzed. The aim of the study was to evaluate recurrence-free survival in patients undergoing transarterial chemoembolization (TACE). The combined model (combining radiomic and clinic-radiological characteristics) showed the best performance for the assessment of relapse-free survival in patients with HCC after TACE. It was possible to divide patients into two different subgroups, high and low risk of relapse [142]. Notably, by employing CEUS-based deep learning radiomics models, Ma et al. attempted to predict early HCC recurrence after ablation and stratifying patients into subgroups at high and low risk of late recurrence [143].

Conversely, the most common hepatic lesions are not primaries, but metastases, which are 18–40 times more frequent than primary tumors [144]. Out of necessity, therefore, several radiomics studies over the years have focused on hepatic secondarisms. Taghavi et al. attempted to predict the presence of metachronous liver metastases from colorectal cancer by studying microvascular changes in healthy liver parenchyma using a radiomic model based on machine learning. It was demonstrated that the combined model (AUC 95%) was better able to predict the development of secondarisms in the 24 months after diagnosis, compared with the clinical model (AUC 71%) and the radiomic model (AUC 86%) [145]. A further study investigated the predictive value of radiomics on the presence of synchronous liver metastases in CRC patients. Classical imaging techniques, such as CT and MRI with mdc, can in fact detect the presence of liver lesions, but they do not always lead to appropriate accuracy and sensitivity [146]. A combined predictive model and normogram was constructed using radiomic features, CEA, and CA19-9 levels [147]. It has also been shown that radiomic characteristics can predict treatment efficacy in patients with hepatic secondarisms [134]. In recent years, two papers have been focused on this. Ravanelli et al. used contrast-enhanced CT to extract radiomic features and showed that texture is able to predict the response to Bevacizumab, and it was the best predictor of both overall survival and disease-free survival in patients with liver metastases from unresectable colorectal cancer. In particular, it has been observed that the uniformity of lesions, which is strictly dependent on angiogenesis, correlates with a worse response to treatment [148]. Nakanishi et al. developed a model to predict the response of liver metastases to first-line oxaliplatin-based chemotherapy in patients with CRC using radiomic features extracted from pretreatment CT scans [149].

However, it must be considered that the quantification of radiomic features can be sensitive to several technical factors, such as CT acquisition parameters (peak X-ray tube voltage and current, slice thickness) and reconstruction parameters. A recent study confirmed the poor technical reproducibility of radiomic models based on CT images in patients with early HCC recurrence [150].

To conclude, in liver cancer, two meta-analyses have been published recently, both claiming that analyzed data are sparse and heterogeneous [134,151].

In conclusion, in liver cancer, two meta-analyses have been recently published, both concluding that analyzed data are sparse and heterogeneous [134,151]; in particular, Fiz et al. [134] included 32 studies in their review and found that entropy and homogeneity were the radiomic features with the strongest clinical impact. Higher entropy at baseline and lower LM homogeneity were associated with better survival and higher chemotherapy response rates. Decreased entropy and increased homogeneity after chemotherapy were correlated with radiological tumor response. Entropy and homogeneity were also highly predictive of the degree of tumor regression. Finally, it was shown that texture analyses could differentiate LM from other liver tumors.

In another meta-analysis Beckers et al., [151] analyzed 16 studies showing that ADC (apparent diffusion coefficient, on MRI) is the most promising predictor of response and survival, whereas in studies related to CT, texture features show promising results. In FDG-PET(-CT), the results were rather ambiguous.

There are still many limitations, such as small sample size, retrospective design, lack of validation datasets, and unavailability of univocal cut-off values of radiomic features. Currently, two prospective studies are recruiting patients to compare imaging findings, genomics, and pathology parameters.

## 6. Texture Analysis and Prognosis—Focus on Rectal Cancer

Rectal cancer (RC) accounts for one third of all colorectal cancers and is one of the leading causes of cancer death in the Western world in both sexes [25,152,153,154]. IN actual fact, neoadjuvant chemo-radiotherapy (nCRT) followed by total meso-rectal excision (TME) is the gold-standard treatment for patients with locally advanced rectal cancer (LARC) [155], as it increases PFS and allows, at the same time, a less invasive surgery with a lower frequency of complications, albeit its effects on overall survival (OS) is still to be proven [156,157]; in a significant percentage of patients, nCRT led to a pathological complete response (pCR) [158,159,160], a concept that has led to new conservative strategies (‘‘watch-and-wait’’ or local excision) that could be attempted in order to avoid invasive surgery with the risk of major complications and a possible worsening of the quality of life [161].

Given these premises, the main clinical challenge is to preoperatively diagnose pCR in patients with LARC after nCRT. Although magnetic resonance imaging (MRI) is the standard imaging technique for local staging and re-evaluation after nCRT in RC [162,163,164,165], its clinical utility in predicting pCR after nCRT is still uncertain. In this scenario, radiomics has emerged as a promising tool and several studies have shown promising results in the prediction and early assessment of response to chemotherapy using MRI [166,167,168]. Following the aim of the review, we will focus on studies investigating the prediction of response and the prognosis of LARC patients.

In particular, studies in the literature have shown that several first-order radiomic features, extracted from T2-weighted (WI) images, are associated with the pCR [169,170].

Horvat et al. [171] indicated that radiomic features extrapolated from post C-RT T2W images could predict pCR. Nie K et al. [172], again, reported the predictive value of pCR based on radiomic features extracted from pre C-RT DWI MRI, with a promising AUC of 0.79. Chen et al. [173] demonstrated that MRI-based radiomics is a sophisticated and non-invasive tool to accurately distinguish recurrent (LR) lesions from (non-recurrent) lesions at the site of anastomosis and, in particular, the combination of multiple sequences in MRI significantly improved its performance. In this study, the ROC curve of model–combination indicated an AUC of 0.864 (validation set), with sensitivity and specificity of 81–82% and 75–86%, respectively, suggesting that model–combination may provide better discrimination performance than individual models (*p* > 0.05). Interestingly, extracting and combining quantitative features from multiple MRI sequences significantly improves model performance, making it more effective.

During C-RT, intra-tumor heterogeneity is dynamic, therefore, radiomic features extracted from single-sequence images (i.e., pre or post C-RT) may overlook tumor change during treatment and have inherent limitations [174]. Therefore, delta-radiomics, which is defined as the change in quantitative radiomic characteristics in a series of longitudinal images in order to detect information about changes in intra-tumor heterogeneity and to adapt therapy, represents a new frontier in radiomics. Wan et al. [175] developed and evaluated the performance of the delta-radiomic model, based on pre- and post C-RT MRI percentage changes, for pCR prediction after nCRT. The developed combined model (using T2WI and DWI) provided the best performance for pCR prediction with AUCs of 0.91 and 0.91 in the training and validation sets, which were superior to those of the mrTRG and delta-radiomic models developed using only T2WI or DWI. Instead, Boldrini et al. [21] investigated the predictive power of delta-radiomic features extrapolated from hybrid 0.35 T magnetic resonance (MR)-guided radiotherapy (MRgRT) in LARC patients undergoing pre-nCRT. This study showed that the variation in three delta features, such as energy, grey level non-uniformity and least axis length, showed a statistically significant association (*p*-value < 0.05), demonstrating a correlation with the cCR.

Other authors have investigated additional techniques of imaging, such as ultrasound (US), which was used to develop a radiomic model based on the machine learning of the US to pre-operatively predict tumor deposits (TD) [176]. Studies have confirmed that TD-positive patients have more aggressive tumors, with poorer disease-free survival (hazard ratio, HR, 1.7 to 2.0) and poorer overall survival (HR, 2.2 to 2.9) [177]. Yuan et al. [178], on the other hand, used a machine learning technique to build radiomics models by extracting radiomic features on non-contrast CT images, and predict the response to treatment. Bibault et al. [179] reported 80% accuracy in predicting a complete response in LARC with nCRT using radiomic features extracted from a post-contrast CT through a deep neural network algorithm. Chee et al. showed that first-order features extracted from CT were associated with response to CRT [180].

Other investigators have investigated the role of TA in the prognosis, with the prediction of distant metastases (DM). A delta-radiomics approach was used to predict the occurrence of distant metastases by Chiloiro et al. [181], whereas another study [182] validated a radiomic model capable of predicting postoperative DM by stratifying patients who might benefit from adjuvant chemotherapy. Liu et al., instead, evaluated the role of pre-nCRT MRI radiomic parameters in predicting synchronous DM in 177 patients with RC with an area under the ROC (AUC) of 0.827 (95% confidence interval (CI), 0.6963–0.9580) [183]. Nardone et al. [184] found a correlation between pre-nCRT MRI texture analysis and early disease progression in 49 patients. In particular, this study showed that patients with a higher GLCM contrast and a lower GLCM correlation showed recurrence and/or DM within three months after radical surgery.

In conclusion, the applications of artificial intelligence in the field of rectal cancer oncology management are varied and very promising, as it could contribute to increasingly personalize patient management and improve survival.

Two meta-analyses have been recently published in the field of colorectal cancer [185,186], with heterogeneous results regarding radiomics methods and included features. At the same time, several studies were able to predict response with good performance. In particular, a recent meta-analysis [186] showed that, at present, few studies have investigated the predictive value of first order radiomic features and that, among them, kurtosis was found to be a significant predictor of treatment response, achieving the highest AUC 0.91 in predicting pCR; for second order radiomic features, more disparate results were obtained with AUCs ranging from 0.54 to 0.99.

Currently, several observational prospective studies are under development with the aim to predict response to neoadjuvant therapies and/or predict cancer prognosis (see Table 1).

## 7. Texture Analysis and Prognosis—Focus on other Cancers

TA has been applied in several settings in many other cancer diseases, such as bladder and prostate cancers, brain cancer (both primary or metastastic), sarcoma, kidney cancer, Hodgkin and non-Hodgkin lymphoma, gynaecological cancer, and head and neck cancers.

These heterogeneous diseases are usually managed with a combination of different strategies, such as surgery, radiotherapy, chemotherapy and immunotherapy. In this context, TA has been applied for different purposes, such as the prediction of prognosis, the correlation with grading or other histological characteristics, and differential diagnosis both in the diagnosis and in the response to therapies.

Many meta-analyses have recently been published in this heterogeneous context and, in this paragraph, we will summarize their results in the above-mentioned diseases.

Kozikowski et al. investigated the role of TA in the prediction of muscle-invasive bladder cancer and analyzed eight studies with a total of 860 included patients [187]. The authors found several differences in approaches, although TA models were found to be relatively homogeneous in diagnostic accuracy.

Stanzione et al. performed a meta-analysis of TA approaches in prostate cancer and assessed the 73 included studies with a radiomic quality score (RQS) [188,189]. The authors concluded that prostate TA still lacks the quality required to allow for its introduction in clinical practice, due to the lack of feature robustness testing strategies and external validation datasets. On the contrary, Castaldo et al. analyzed radiomic and genomic machine learning methods in the detection of prostate cancer [190] and found that, despite the above limitations, the performance was considered satisfactory for several studies investigating multiparametric magnetic resonance imaging and urine biomarkers.

In brain cancer, several TA studies have been published in the last decade, mainly in the field of grading prediction of glioma and meningioma, prognosis prediction and differential diagnosis of progression versus radionecrosis in radiotherapy-treated patients. Tabatabaei et al. and Ugga et al. recently performed similar meta-analyses for the prediction of grading in glioma and meningioma, respectively [191,192]. The authors both concluded that future studies with adequate standardization and higher methodological quality are required, prior to the introduction of TA in clinical practice for this purpose. The same conclusion was reached by Kim et al. in the setting of TA prediction of true progression versus radionecrosis after radiotherapy for brain metastases [193].

Crombè et al. explored the potential of TA in different sarcoma diseases [194] and included 52 studies. They concluded that, despite the promising results, further efforts are needed to make sarcoma radiomics studies reproducible with an acceptable level of evidence. In this context, Gitto et al. investigated the reproducibility and validation strategies, and found a huge variation among different studies, thus mining the introduction of TA in the clinical setting [195].

Different TA approaches have been used in the setting of renal cancer, mainly with the aim of differential diagnosis of histological subtypes in the prediction of therapy response and survival. Ursprung et al. calculated an odds ratio of 6.24 (95% CI 4.27–9.12; *p* < 0.001) for the differentiation of angiomyolipoma without visible fat from renal cell carcinoma [196]. Mühlbauer et al., similarly, concluded that this approach seems promising in the differential diagnosis of histological subtypes, but shared data and open science must aid in improving reproducibility of future studies [197].

Frood et al. investigated the role of baseline nuclear medicine imaging in the prediction of treatment outcome in Hodgkin and diffuse large B cell lymphoma [198] and concluded that further work in harmonization, segmentation and performance cut-off is required to develop robust methodologies that are amenable for clinical utility.

TA approaches have been tested in gynaecological cancers with different aims, such as the prediction of prognosis of ovarian cancer patients [199] or the radiological preoperative assessment of patients with endometrial carcinoma [200]. There is currently insufficient evidence on the benefit of TA approaches in this context, despite this field being promising for future clinical practice.

TA approaches have also been used in the field of head and neck cancers, mainly with the aim of predicting radiotherapy side effects or to predict prognosis. Carbonara et al. included 8 papers in their meta-analysis, presenting data on parotid glands, cochlea, masticatory muscles, and white brain matter after head and neck radiotherapy [201]. Unfortunately, the RQS of presented studies was low and further studies are needed in the future to validate the TA in this setting. Creff et al. and Guha et al. analyzed the potential role of TA in the prediction of prognosis [202,203] and both authors found significant heterogeneity among the included studies, with the lack of robust external validation studies on the reproducibility and accuracy of TA.

## 8. Conclusions

All the above-mentioned studies have shown interesting results or texture analysis approaches in different diseases. At the same time, there are still several pitfalls to resolve before TA can be successfully applied in the clinical management of cancer patients.

The standardization of both image acquisition and feature extractions, as well as data sharing and the distrust of the clinicians in the black box approach represent the major problems to solve in the future.

The ideal solution for standardization is to define the methods in dedicated prospective trials. It is necessary to underline the fact that, despite an impressive number of retrospective published papers, the number of prospective trials investigating radiomics in cancer therapy is inexplicably low.

Additionally, several efforts should be made in the development of high-number and high-quality shared databases in the future. These datasets require joint efforts by both companies and institutions, such as Cancer Learning Intelligence Network for Quality and Flatiron Health, the Cancer Imaging Archive.

Radiomic parameters are very difficult to describe and to refer to known clinical variables, thus TA looks like a black box to the Clinicians. In this regard, further work should be performed to correlate radiomics to underlying clinical and molecular connotations, with approaches such as radiogenomics.

At the same time, the use of structured reports in radiology in the future could be integrated with radiomic data correlated to prognosis or response to therapies, in order to facilitate the clinicians to understand the TA approach [204,205,206].

By setting up the ambitious goal of addressing these challenges, radiomics can become a clinical reality in the next few years and a higher number of prospective trials can be designed and conducted in the near future.

## Figures and Tables

**Table 1 diagnostics-11-01796-t001:** Registered Clinical trials investigating Radiomics approach in Oncology.

NCT Number	Study Phase	Disease Stage	Trial Design
NCT03709186	Observational	Breast Cancer	Evaluation of radiomic markers in breast tumors to predict metastatic risk based on radiomic features following primary therapy with imaging techniques (DCE-MRI and DWI-MRI).
NCT03038568	Observational	Colorectal Cancer Metastatic	Measurements of tumor differences vary with slight changes in CT scan parameters. Reproducible radiomic features can be extracted for abdominal tumors, and specifically colorectal liver metastases.
NCT01585545	Observational	Lung Cancer	To evaluate relationships between multiparametric imaging biomarkers (CT, PET/CT, MRI) and genetic analysis in NSCLC patients.
NCT04315753	Observational	Lung Cancer	Analysis of the role of molecular and cellular biomarkers (exosomes antigens, circulating tumor cells-CTCs, panel of mutations in circulating free DNA) and radiomic signature.
NCT04323579	Observational	Lung Cancer	CLEARLY will focus on validation of a multifactorial “bio-radiomic” protocol for early diagnosis of lung cancer that combines circulating biomarkers and radiomic analysis.
NCT04364776	Observational	Lung Cancer	An Observational Study on Computed Tomography as an Image-based Predictive Marker of Response to Chemoradiation Followed by Durvalumab in Stage III Unresectable Non-small Cell Lung Cancer (NSCLC).
NCT03787667	Observational	Colorectal Cancer	This study is to collect and analyze data of radiomics (on enhanced CT or MRI) of primary site or metastasis of colorectal cancer aiming to make a precise preoperative diagnosis and long-term prognosis evaluation.
NCT03872362	Observational	Lung Cancer	The project aims to develop and validate radiomics models based on CT images to identify malignant nodules and then to discriminate the different types of lung adenocarcinoma in patients with pulmonary nodules.
NCT03198975	Observational	Hepatocellular Carcinoma	The aim of this prospective study is to develop a machine learning-based model for preoperative prediction of MVI by extracting high-dimensional magnetic resonance (MR) image features.
NCT03679936	Observational	Lung Cancer	The purpose of this study is correlate the imaging findings with genomics and histopathological features of newly diagnosed non-small cell lung cancer (NSCLC).
NCT04457700	Observational	Breast Cancer	This study aimed to assess the performance of CT-based radiomics in evaluating the response and predicting pCR of metastatic lymph nodes after NAC in breast cancer patients.
NCT04553601	Interventional	Lung Cancer	To assess the potential usefulness of radiogenomics for tumor driving genes heterogeneity in non-small cell lung cancer.
NCT04320030	Interventional	Breast Cancer	This phase II study is assessing the correlation between M1/M2 macrophage polarization determined by tumor immunohistochemistry analysis and [18F]DPA-714 PET/CT binding (qualitative and texture analysis) in patients with operable triple negative breast cancer.
NCT01959490	Interventional	Breast Cancer	Next Generation Sequencing and radiomics to Evaluate Breast Cancer Subtypes and Genomic Predictors of Response to Therapy in the Preoperative Setting for Stage II-III Breast Cancer.
NCT02439086	Interventional	Rectal Cancer	Prediction of complete response in PET and MRI texture analysis.

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
