# Peer review of "Radiomics as a New Frontier of Imaging for Cancer Prognosis: A Narrative Review"

_diagnostics, 2021, doi:10.3390/diagnostics11101796_

Round 1
Reviewer 1 Report
I have read with interest this manuscript. The paper is stimulating and it was a pleasure to review it, but I would like to ask the authors to perform a language editing to make it more fluent to read.
I have some comments concerning the manuscript
- The Authors evaluated the role of texture analysis in the prediction of response to therapy and in stratification of prognosis in patients with cancer. They focused their attention on lung, gastric, rectal, breast and liver cancer. Please comment on this choice, arguing why these specific cancers were selected.
- In the sections “Texture analysis and prognosis”, the succession of short sentences listing the aim of the various papers, without connection between them, makes parts of the text difficult to follow. In addition, a summary of the significant results reported by previous papers could be inserted at the end of each section, to clarify the state of current situation.
Author Response
We thank the Reviewer for his/her effort reading and evaluating our work. We believe that thanks to his/her suggestions our manuscript has improved.
Point 1) We have chosen the above mentioned diseases as in the field of radiomics these diseases were better studied. We have added another paragraph, following the Reviewer 2 suggestion, with a general summary of radiomics inà other cancer diseases.
Point 2) We have performed an English Revision and we have added a summary of the significant results in each section.
Reviewer 2 Report
The authors provide a good review for Textual analysis and prognosis in key cancer. This review provides an overall wide-range view of the impact and the advantages that TA has had in cancer detection since its development.
My only comment for the author is just like the focus was provided for select cancer in 2 to 5, can this be done for some key cancers? or alternatively provide one more section that covers all the other cancers not highlighted in 2-5? A more generic review across other cancer types mentioned in Table 1 but with no section.
Author Response
We thank the Reviewer for his/her effort reading and evaluating our work. We believe that thanks to his/her suggestions our manuscript has improved.
Point 1) We have added another paragraph, following the Reviewer’ suggestion, with a general summary of radiomics in other cancer diseases.
Round 2
Reviewer 1 Report
The paper appears more exhaustive after the changes made by the Authors. I have a suggestion: The word "conversely" is repeated several times in the text. Please, reduce it use.
Author Response
We thank the Reviewer 1 again for his/her efforts evaluating our manuscript. We have replaced the term with synonyms.
Reviewer 2 Report
The authors addressed my comments.
Author Response
We thank the Reviewer again for his/her efforts evaluating our manuscript.